# Expanding the Disorder-Function Paradigm in the C-Terminal Tails of Erbbs

**DOI:** 10.3390/biom11111690

**Published:** 2021-11-14

**Authors:** Louise Pinet, Nadine Assrir, Carine van Heijenoort

**Affiliations:** Institut de Chimie des Substances Naturelles, CNRS UPR2301, Université Paris-Saclay, 91190 Gif-sur-Yvette, France; nadine.assrir@cnrs.fr (N.A.); carine.van-heijenoort@cnrs.fr (C.v.H.)

**Keywords:** intrinsic disorder, signal transduction, receptor tyrosine kinases, ErbB, EGFR, HER

## Abstract

ErbBs are receptor tyrosine kinases involved not only in development, but also in a wide variety of diseases, particularly cancer. Their extracellular, transmembrane, juxtamembrane, and kinase folded domains were described extensively over the past 20 years, structurally and functionally. However, their whole C-terminal tails (CTs) following the kinase domain were only described at atomic resolution in the last 4 years. They were shown to be intrinsically disordered. The CTs are known to be tyrosine-phosphorylated when the activated homo- or hetero-dimers of ErbBs are formed. Their phosphorylation triggers interaction with phosphotyrosine binding (PTB) or Src Homology 2 (SH2) domains and activates several signaling pathways controling cellular motility, proliferation, adhesion, and apoptosis. Beyond this passive role of phosphorylated domain and site display for partners, recent structural and function studies unveiled active roles in regulation of phosphorylation and interaction: the CT regulates activity of the kinase domain; different phosphorylation states have different compaction levels, potentially modulating the succession of phosphorylation events; and prolines have an important role in structure, dynamics, and possibly regulatory interactions. Here, we review both the canonical role of the disordered CT domains of ErbBs as phosphotyrosine display domains and the recent findings that expand the known range of their regulation functions linked to specific structural and dynamic features.

## 1. Introduction

Receptor tyrosine kinases (RTKs) are key players in cell signaling, and are essential for cell proliferation, motility, differentiation, and apoptosis [1]. ErbBs form one out of 20 classes of RTKs, and were the first one to be identified: in 1974 [2] ErbB1 was identified as the receptor for the epidermal growth factor (EGF), and was later shown to have tyrosine kinase activity [3]. ErbBs are often referred to as the HER (Human Epidermal growth factor Receptor) or EGFR (Epidermal Growth Factor Receptor) family. It contains four members (the most common names, mainly used in this review, are in bold): **ErbB1/EGFR**/HER1, **ErbB2**/neu/HER2, **ErbB3**/HER3, **ErbB4**/HER4. ErbB receptors ensure a wide range of roles in development and adult organ function [4,5], and knocking out any of them is lethal at early embryonic or perinatal stage [6]. At the cellular level, ErbBs are mitogenic [7,8], but are also involved in cell adhesion [9], motility [10,11], and differentiation [8,12]. They were shown to perform these functions by activating (mainly) the MAP-kinase, PI3K/Akt, Src kinase, STAT5 and MEMO pathways [11,13]. Misregulation and mutations of ErbB proteins were linked very early to cancer [14,15], and EGFR and ErbB2 in particular are now commonly targeted by tyrosine kinase inhibitors and/or antibodies [16]. Additionally, ErbBs are associated with neurodegeneration and especially Alzheimer’s disease [17], as well as with cardiovascular diseases [18].

ErbBs, like all RTKs, are made up of an extracellular domain (ECD) that generally binds ligands, a transmembrane helix (TMH), an intracellular juxtamembrane region (JM), a tyrosine kinase domain (TKD), and a C-terminal tail (CT), sometimes called regulatory domain, that undergoes autophosphorylation of its tyrosines to interact with downstream partners [19]. This structural organization is presented Figure 1. All domains except the long (on average 300 residues) C-terminal tails of ErbBs (CT-ErbBs) were extensively structurally described. As phosphorylation domains involved in signaling, CT-ErbBs can be expected to be intrinsically disordered. Intrinsically disordered proteins (IDPs) are a class of protein that lack a unique, stable three-dimensional structure, and are better described by large conformational ensembles. This favors high accessibility to post-translational modifications, especially phosphorylation [20,21], multiplicity of relatively low-affinity binding with high turnover [22], and regulation mechanisms extending the concept of allostery [23]. IDPs are therefore ideal candidates to participate in signal transduction [24,25]. Recent studies have explicitly confirmed that CT-ErbBs are disordered [26,27,28,29,30]. For a long time, the conformational properties of these tails were not included in studies, and they were treated as mere “strings” attaching the tyrosine phosphorylation sites to the rest of the receptor. However, the growing evidence that the CTs have roles in kinase activity regulation [31,32,33], and that the conformational modulation by phosphorylation may have functional impacts [28,29,34,35] calls for a better structural description.

In this review, we first contextualize the role of the CT of ErbBs in the different steps of signal transduction, and review their role as phosphorylation and interaction domains. We then review structural and functional information that was gathered these last few years that revisit their role beyond passive phosphotyrosine anchors, as active regulators of signal transduction. Finally, we compare CTs of ErbBs to the tails of other RTKs, showing their specificities and possible common regulation functions.

## 2. General Structure and Mechanism of Erbbs

### 2.1. Similarities and Differences between ErbBs

EGFR is often used as the canonical ErbB, and ErbB2 is also widely studied due to its connection to breast cancer [36]. Despite the common domain organization of all four ErbBs, functional differences exist between them and are shown in Figure 1A. Most strikingly, while the activation of most RTKs requires receptor-ligand interaction and autophosphorylation, no ligand was found for ErbB2, and ErbB3 kinase was shown to be very weakly active (about 1,000-fold less active than EGFR) [37]. This coincides with an insert in ErbB2 extracellular domain, next to the ligand binding site, compared to that of other ErbBs, and with a lower degree of similarity of the kinase of ErbB3 to the kinases of ErbB1/2/4 [38].

From the evolutionary point of view, it was found that in invertebrates there is only one ErbB-like protein. Following three steps of gene duplication (separating ErbB1/2 from ErbB3/4 first, with then another gene duplication of each), the four distinct ErbBs appeared in vertebrates [39]. Although EGFR and ErbB2 have different functional specificities (ligand-binding properties, signaling pathways etc.), ErbB2 is therefore the closest homolog of EGFR (the most-studied ErbB) in the family. ErbB3 and ErbB4 are less studied: as of 30 August 2021, a PDB search for (“ErbB3” OR “HER3” OR “ErbB4” OR “HER4”) gave 112 results as opposed to 1210 results for (“ErbB1” OR “HER1” OR “EGFR” OR “ErbB2” OR “HER2”). ErbB3/4 mechanisms may therefore differ slightly from the general mechanism described in this section.

The numbering used in this review is given in Figure 1D for EGFR and ErbB2, the most studied ErbBs.

### 2.2. The Different Steps of Activation, from Extracellular Ligand Binding to Kinase Activation

Given their implication in both important physiological functions and cancer, a lot of effort was put into understanding the mechanism of signal transduction by ErbBs. Structural studies were key to this understanding, and each step of signal transduction down to the kinase domain was determined or confirmed with crystal and/or NMR structures. No high-resolution structure of a full-length ErbB exists, so the series of steps were inferred from fragments of the receptors and connected to functional data. A full-length model, lacking only the CT domain, was reconstituted by Jura et al. [31], and is shown in Figure 1C.

Briefly, ligand binding to ErbB1, ErbB3 or ErbB4 ECD is followed by a conformational change that enables formation of active homo- or hetero-dimers of the receptors. ErbB2, on the contrary, does not bind any known ligand and is constitutively poised to dimerize. Dimerization enables the activation of the intracellular TKD of at least one monomer of the dimer and then leads to tyrosine phosphorylation of the CT. Finally, the phosphorylated tails serve as anchors for interaction with adaptor proteins that trigger signaling. A schematic view of the steps is given in Figure 1B.

Understanding the general mechanism of signal transduction by ErbB receptors is crucial to understand the role of their C-terminal tails, their different states (signaling-potent or not), and their direct or indirect interactions with the other domains. Therefore, we briefly review all the steps below, before focusing on the CT. A more extensive review of these steps was written by Lemmon, Schlessinger, and Ferguson [40]. We discuss how the conformation, and therefore activation state, of each domain is dependent on the other domains of the receptor.

#### 2.2.1. Ligand Binding to the ECD and Dimerization

ErbBs ECD is composed of about 630 residues divided into four domains (see Figure 1B,D). Crystal structures of EGF-bound [41] and free [42] ECD of EGFR revealed the first step in ErbB dimerization (Figure 1B). As deciphered by Ferguson, Lemmon and colleagues [42], pioneers in the structure-function description of ErbBs, in the free, autoinhibited state, the dimerization site (in domain II) is masked by interaction with domain IV. Upon ligand binding, mainly to domains I and III, a large conformational change frees domain II from intramolecular interaction with domain IV, and permits dimerization of the ECD. For each of the ErbB1/ErbB3/ErbB4 receptors, binding of different ligands leads to different types of dimers and different dimer association constants, with different signaling specificities [43,44,45].

In addition to this ligand-induced dimerization model, a ligand-induced activation of inactive dimers was also proposed [46,47], in the so-called “rotation model”, reviewed by Purba et al. [48]. Whether the monomers or inactive dimers predominate might depend on the nature of the receptor, level of expression, and localization in the membrane, since membrane composition is not homogeneous [49]. Structural studies of the ECD were performed on monomers and homodimers. No structure of heterodimeric ECD of ErbB receptors was, to our knowledge, ever described, because isolated ECDs homodimerize preferentially [50]. Heterodimerization is driven by other regions of the receptor as described below.

#### 2.2.2. The Transmembrane Helix (TMH) and N-Terminal End of the Juxtamembrane Region (JMA)

Yarden and Schlessinger, two key investigators of ErbBs since their discovery, determined that dimerization of the ECD itself is not sufficient for ErbB receptors signal transduction, and that other intermolecular interactions are required [51]. In particular, the TMH was shown to enhance ligand-induced homodimerization of the ECD [52]. It is now even considered that the ECD dimerization per se is not needed, since truncated forms of ErbBs lacking the ECD are active [14], and isolated TMH homo- and hetero-dimerize [53,54,55,56].

Furthermore, mutations in the TMH are involved in overactivation of ErbBs and diseases [57]. The origin of activation of the receptor by the TMH was suggested to be due to rotation of the TMH to form a hydrogen bond with the other TMH of the dimer, triggering the formation of an antiparallel dimer in the JMA region (see Figure 1C) [58,59,60,61,62]. Conversely, the suggested model for the inactive state shows a membrane embedded, parallel, JMA dimer [59]. In line with this model, Arkhipov et al. showed that the interaction of the intracellular part of ErbBs with the plasma membrane is inhibitory [63]. It was also shown that the composition of the membrane and its fluidity, that are sometimes altered in diseases, regulate the dimerization of ErbBs [64].

#### 2.2.3. The Tyrosine Kinase Domain and Its Activation

In many RTKs, monomeric kinase activity is low. Dimerization of the receptors causes *trans*-tyrosine phosphorylation of the kinase or juxtamembrane domains, which dramatically increases kinase activity by changing the conformation of the activation loop [65]. However, Tyr to Phe mutagenesis studies on ErbBs [66] showed that optimal kinase activity in the dimers does not require phosphorylation of this tyrosine (Y869 in EGFR) in the activation loop, and the crystal structure of EGFR unphosphorylated kinase domain [67] showed a conformation typical of the active state of a tyrosine kinase (right structure of Figure 2A). An EGFR kinase structure resembling the inactive, unphosphorylated state of the kinase of other RTKs was described two years later, bound to a a tyrosine kinase inhibitor (Lapatinib) [68].

A closer look at the crystal lattice of the active EGFR [67], and mutagenesis studies [62] showed that the active form is an asymmetric dimer as schematically shown in Figure 2B. This dimer is similar to the CDK/cyclin complex and not to other RTK kinase domains: one kinase monomer plays the role of an activator, promoting through intermolecular interactions the activation of the other, receiver kinase.

The mechanisms determining the relative position of each kinase (receiver/activator) is not clear. It was proposed that it is dependent on the binding of ligand to the ECD of each monomer [69], but other studies suggested that it is intrinsic to the composition of the dimer, each receptor having a certain propensity to be the receiver (and EGFR being the “best receiver”) [70].

The C-terminal half of the juxtamembrane region (JMB) was shown to be important for kinase activity [71], through the interaction of the JMB segment of the receiver with the C-lobe of the activator, locking the active dimeric conformation (Figure 2B) [31,60].

From the observation that ErbB3 tail can be phosphorylated even though it is kinase dead, and other experiments with kinase-dead variants [72], *trans*-autophosphorylation (phosphorylation of the tail of one ErbB monomer by the kinase of the other ErbB of the dimer) is efficient. The coexistence of *trans*-autophosphorylation with *cis*-autophosphorylation (phosphorylation by its very own kinase) was, however, never ruled out. The prevalence of *trans*-autophosphorylation is also consistent with the sightly higher level of phosphorylation in the tail of the activator monomer compared to that of the tail of the receiver monomer in an asymmetric dimer [33].

All the conformational changes from the extracellular domains to the kinase domain eventually lead to phosphorylation of the CT. This is the final and decisive step that triggers the interaction with different partners, each starting a different signaling pathway and determining cell fate.

### 2.3. The C-Terminal Tail (CT) and Signal Transduction by Phosphotyrosines

#### 2.3.1. Tyrosine Autophosphorylation Sites

At the end of the chain, the role of the CT in ErbB receptors is to ensure signal transduction through interactions of phosphorylated tyrosines with signaling proteins (containing mainly SH2 and PTB domains) that are shown in Figure 3B. These tyrosines are called autophosphorylation sites when they are phosphorylated by ErbB kinases in the dimer, as opposed to tyrosines phosphorylated only by other tyrosine kinases in the cell (such as the Abelson kinase or Src kinase [73,74,75]). There are 9 tyrosines in CT-EGFR, amongst which six can be autophosphorylated. Four of them were shown to be highly (i.e., to a high percentage) autophosphorylated sites (Y1092, Y1110, Y1172 and Y1197), and two have lower levels of phosphorylation (Y1016 and Y1069) [76,77,78]. Phosphorylation studies of EGFR were reviewed by [79], and an overview table is shown in Figure 3A. It was suggested that phosphorylation does not happen at many sites at once [80], but phosphorylation mechanism and patterns are widely unknown in the actual signaling context.

Similarly, five autophosphorylation sites (one containing two adjacent tyrosines) out of 9 tyrosines were identified in ErbB2 [81,82]: Y1023 (called Y${}_{A}$) Y1139 (Y${}_{B}$), Y1196 (Y${}_{C}$), Y1221/1222 (Y${}_{D1}$ and Y${}_{D2}$) and Y1248 (Y${}_{E}$). For ErbB3 and ErbB4, the number of autophosphorylation sites is unclear. For all ErbBs, which sites will actually be phosphorylated, and therefore which signaling pathways will be activated, depends on the potential ligand bound and on the dimerization partner [83].

#### 2.3.2. Partners of Phosphorylated Tyrosines of ErbBs

The phosphotyrosine interaction sites of ErbBs include autophosphorylation sites, but also some tyrosines that can be phosphorylated by other kinases. Partners of all these ErbB phosphotyrosines include Grb2 and Shc (MAP kinase and Akt pathways), PI3K (Akt pathway), Src and STAT5 (MAPK and Akt pathways) [13,84]. In particular, Schulze et al. [84] identified ErbB partners in HeLa cells by pull-down coupled to mass spectrometry (see Figure 3B). Some partners are common between different ErbBs and some are not. Grb2 and Shc are the adaptor proteins that were reported to interact with all ErbBs. The signaling interaction network of ErbBs was reviewed by Yarden and Sliwkowski [13], with the main addition to the knowledge of 2001 being the discovery of MEMO (Mediator of ErbB2-driven cell MOtility) [11]. This protein, homologous to iron dioxygenases [85], binds phosphotyrosine 1222 of ErbB2 [86] and controls microtubule and actin networks [87].

The few experimental structures that exist of proteins bound to ErbB phosphotyrosine peptides are shown in Figure 4 (top row). Phospholipase C-γ1 (PLCG1) and Protein Tyrosine Phosphatase Non-receptor type 18 (PTPN18) structures are crystal structures, while Grb7 was studied by solution NMR. In all the structures, the ErbB peptides are extended, except for the complex with Grb7. In this structure, similar to structures of Grb2 with ligand peptides, the peptide forms a β-turn at the C-terminus of the phosphotyrosine. This ligand conformation was shown to be a specificity of Grb proteins, and not of CT-ErbBs [88,89].

#### 2.3.3. Shortcuts and Pitfall of the Common Signal Transduction Model by CTs

In most studies, molecular details of the interactions of CT-ErbBs with downstream partners are studied using short peptides of ErbBs (see Figure 4). This leaves out possible long-range conformational effects and multisite binding, and can bias our understanding of these interactions. Just like ErbBs structured domains are interdependent and allosterically communicate with each other, the different tyrosine sites in CT-ErbBs may communicate, and the CT itself may communicate with other domains of the receptors. Due to the disordered nature of the CTs, this aspect was long neglected, and time has come to investigate CT-ErbBs as thoroughly as folded domains to unveil their regulation potential.

## 3. CTs of ErbBs, More Than Disordered Ropes with pY Anchors

From the role of the CTs described above, as phosphorylation and interaction domains, they may be classified as “display sites”, as defined in the functional classification by van der Lee et al. [90]: “flexibility of IDRs facilitates exposure of motifs and easy access for proteins that introduce and read PTMs”. This does not exclude the presence of additional mechanisms linked to disorder and phosphorylation. Phosphorylation was shown, in some systems, to significantly modify structure and function [91], with changes as dramatic as folding upon phosphorylation for 4E-BP2 [92]. Even more subtle conformational changes upon phosphorylation can lead to change from an auto-inhibited state to an active state (as in CFTR R region [93]) or vice-versa (in p53 TAD [94]). IDRs were also shown to be important for kinase regulation in signaling networks [95]. Even though mutations in the CT of ErbBs are much less commonly linked to disease compared to mutations in the kinase domain [96], which is expected for a disordered region, CT-deletion mutants were linked to constitutive receptor activation and glioblastoma [97], which underlines the importance of that region for regulation.

### 3.1. Sequence Characteristics of the Four CT-ErbBs

Contrary to the extracellular and kinase domains, the C-terminal tails are poorly conserved amongst the four ErbB receptors: only 11 to 25% identity, compared to 59 to 81% identity for the kinase domain [84]. However, from a simple sequence analysis and the use of predictors, common characteristics and differences emerge, as seen in Table 1. The first observation is that the tail of all ErbBs is predicted to be disordered, although CT-ErbB2 and CT-ErbB3 are predicted to be more disordered than CT-ErbB1 and CT-ErbB4. Disorder usually arises from a low percentage of hydrophobic residues and from an enrichment in charged, small, polar residues and/or prolines [98]. Comparing the content of some of these amino acids between ErbBs and with different sets of proteins can give some insights about the origin of disorder in them.

The difference of disorder prediction for ErbB2/ErbB3 compared to ErbB1/ErbB4 correlates with a significantly higher glycine content in the latter pair, which probably affects flexibility substantially. The other parameters we looked at are in favor of an extended conformation of the chain. All ErbBs have a very high content in proline, the most disorder-promoting amino acid. Additionally, electrostatic repulsion might arise from the charge unbalance in favor of negatively charged amino acids, with a striking depletion in lysines, resulting in negative net charge and low pI values. Finally, CT-ErbBs are rich in tyrosines, which is expected from their nature of RTKs but relatively rare in IDPs, in which most commonly phosphorylated residues are serines and threonines.

### 3.2. The Need for Tight Regulation: The Specificity of Tyrosine Phosphorylation Sites

Tyrosines were shown to make up less than 1% of all phosphorylated residues in a cell [101]. This is due both to the activation of tyrosine kinases in very specific conditions, and to the high turnover of tyrosine phosphatases, making tyrosine phosphorylation a very regulated process compared to serine/threonine phosphorylation [102]. Accordingly, it was shown that many tyrosine kinases are receptors: amongst the 90 tyrosine kinases identified in the human genome, 58 are RTKs [103]. Tyrosine phosphorylation was shown not to be strongly sequence-specific, even less so than serine/threonine phosphorylation, but rather context-specific (local concentration, accessibility) [104]. It is thus expected that regulation processes in ErbBs, like in other RTKs, are essential. This regulation can notably be kinase activity regulation, or conformational regulation modulating either local accessibility of the tyrosine site or topological accessibility of the kinase domain to the tyrosine site. Additionally, regulation of the whole receptor using internalization can intervene. These regulation mechanisms can be either intramolecular or intermolecular within the signaling dimers, and can also involve partners other than the signaling ones. In this section, we review some studies focusing on the role of CT-ErbBs in these regulation mechanisms.

### 3.3. Regulatory Roles of CT-ErbBs

#### 3.3.1. Interdependence of Phosphorylation Sites

An interesting study of the interdependence of phosphorylation sites was performed by Dankort et al., in the laboratory of Muller, on neu (the rat ErbB2). They studied in depth the autophosphorylation sites of neu [105,106,107]. Five autophosphorylation sites (of which one “double” site with two adjacent tyrosines) were identified [81,82]: Y1023 (called Y${}_{A}$) Y1139 (Y${}_{B}$), Y1196 (Y${}_{C}$), Y1221/1222 (Y${}_{D1}$ and Y${}_{D2}$) and Y1248 (Y${}_{E}$). The authors studied the transformation potency, signaling pathways and interaction partners of each of them, using tyrosine-to-phenylalanine mutations [105,106,107]. They showed that individual phosphorylation of Y${}_{A}$ inhibited signal transduction, contrary to all other sites that were individually sufficient for transformation. Phosphorylation of Y${}_{B}$, Y${}_{C}$ or Y${}_{E}$ did not abolish Y${}_{A}$ inhibition, whereas phosphorylation of Y${}_{D}$ did. The inhibitory role of the corresponding tyrosine of EGFR (Y998) was also previously observed [108]. These data suggest a mechanism of regulation that is more complex than just direct phosphorylation-interaction-signaling, and may involve conformational regulation.

#### 3.3.2. Regulation of Kinase Activity by the CT

The CT is believed to participate in the regulation of the activity of the kinase domain. Deletion of some parts of the CT modulates kinase activity. However, the results of the different studies [78,97,109,110,111] diverge on the nature and precise regions of the tail involved in this regulation. This regulation could be phosphorylation-dependent, creating feedback loops. Indeed, tyrosine to phenylalanine mutants in the CT of activated EGFR showed different kinase activities [109,112]. Both mutation and deletion experiments are difficult to interpret in light of the possible cross-talk between different tyrosine phosphorylation and interactions; structural biology is a major help in that regard.

Crystal structures of EGFR (Figure 2A) show the implication of the CT in the kinase domain conformational state, supporting a regulatory function. In the inactive dimer [31], a helix is present at the beginning of the CT. This helix (991ȓ1002) is sometimes called the AP-2 helix from its interaction with the clathrin-associated protein complex AP-2. It interacts with parts of the kinase N-lobe close to the N-lobe-C-lobe hinge region and ATP-binding site of both monomers, which could alter activity. This is consistent with the observation that C-terminal deletion increases ATP-binding of the kinase domain [111]. Autophosphorylation of the intracellular variant partially disrupts the contact between the CT and ATP-binding site [34].

The AP-2 helix is followed by an extended, negatively charged part (1003–1014) called the “electrostatic hook” that interacts with the C-lobe of the same monomer in the inactive conformation. There is therefore a competition, shown in Figure 2B, of this hook with the JMB segment for interaction with the C-lobe, the JMB-C-lobe contact being observed in the active asymmetric dimer [62]. In the active conformation the hook interacts with the N-lobe, leaving space for the JMB-C-lobe interaction. Further down the C-terminus, the ${}^{1034}$LLSSL${}^{1038}$ motif is suggested to form a β-sheet completing a strand of the C-lobe of the kinase [33]. Molecular dynamics studies [113], as well as sequence conservation studies across ErbBs [32] support the competition between the C-terminal tail and JMB for kinase regulation in the family. Interestingly, an important residue for the contact between the kinase domain and the C-terminal tail, residue 1016, is a phosphorylatable tyrosine in most ErbBs, and an aspartate in ErbB3 [32]. The negative charge of phosphate or of the aspartate side chain may disrupt the contact and explain the modified kinase activities in the phosphorylated form and in ErbB3.

From structural data, the first part of the tail seems to prevent formation of the active asymmetric dimer, which is consistent with the study by Pines et al. that shows that deletion of the first 70 to 80 residues of the CT leads to constitutive dimerization, increase in kinase activity and activation of signaling pathways [97]. These deletions correspond to mutations observed in glioblastoma (EGFR vIVa and vIVb). Smaller deletions also show that the auto-inhibition is mainly but not only due to the region corresponding to the AP-2 helix and electrostatic hook, and extends further towards the C-terminus, up to residue 1075 [33]. Closer to the C-terminus, the tail has an opposite role, especially Y1110 and its surroundings, and its deletion decreases phosphorylation of the CT. Kovacs et al. showed that it was correlated with the phosphorylation of the activation loop (Y869), that is not essential for kinase activation but could still have a role in slight modulation of kinase activity [33].

A study using EGFR-ErbB3 chimera demonstrated that the inhibition of the kinase was C-terminal sequence dependent, since ErbB3 tail grafted on ErbB1 kinase domain did not induce kinase inhibition, contrary to the nonchimeric kinase-CT variants [114].

#### 3.3.3. The CT Is Involved in ErbB Receptor Trafficking

The C-terminal tail of ErbBs was shown to have a crucial role in the receptor internalization for recycling or degradation, since deleting it [115] or creating chimeras between different ErbBs [116] completely changed internalization levels. Internalization levels and mechanisms, as well as the role of the CT seem to be different in different ErbBs. Although it was initially thought that EGFR was the only one to have active endocytosis pathways [117], it was shown that ErbB3 [118] and ErbB4 [119] also undergo endocytosis, while ErbB2 is only recycled [120,121].

These internalization mechanisms are linked to the sequence and structure of ErbB tails. A prediction of Eukaryotic Linear Motifs (ELMs, elm.eu.org [122], accessed on 2 November 2021) for each ErbB gives four motifs in addition to phosphotyrosine sites: the AP-2 helix site in EGFR, the ErbB2 interacting protein (Erbin) binding site in ErbB2, a CDK phosphorylation motif in ErbB3, and a PPAY motif in ErbB4. These four motifs are all linked to phenomena leading to endocytosis, as discussed below.

As previously mentioned, the helix identified at the N-terminus of the tail is called the AP-2 helix because of its identified interaction with the AP-2 (adaptor protein 2) complex [123] that is responsible for rapid internalization of EGFR by clathrin-coated pits (CCPs), even though AP-2-independent endocytosis pathways also exist [124]. Moreover, Erbin stabilizes ErbB2 by binding to its extreme C-terminus and preventing its degradation [125]. The structures of ErbB peptides with the AP-2 clathrin adaptor and Erbin are shown in Figure 4 (bottom row). In those crystal structures, the ErbB peptides are once again in extended conformations.

The phosphorylation of serines and threonines of CT-EGFR was linked to receptor trafficking. Tong et al. studied phosphorylation patterns in model systems reproducing the conditions after EGF stimulation and before or after internalization. They showed that many serines/threonines were highly phosphorylated after EGF stimulation and EGFR internalization, or after stress induced by anisomycin [126]. In particular, phosphorylation of S991 is necessary for down-regulation of EGFR, and is linked to p38 MAPK-dependent phosphorylation of S1039 and T1041 [127]. These phosphorylations are involved in AP-2 complex interaction, ubiquitination, and downregulation of EGFR [126]. However, there is, to the best of our knowledge, no extensive analysis of the mechanisms of phosphorylation and of the molecular details leading to endocytosis processes. For example, the relevance of the CDK site of ErbB3 identified by ELM prediction remains to be investigated.

Phosphotyrosine sites are also involved in receptor trafficking, for example with Grb2 binding the tail of EGFR, recruiting Cbl for ubiquitination and subsequent internalization in CCPs [128]. Finally, some proline motifs are involved in ErbB trafficking: the PPAY motif identified in ErbB4 was shown to allow binding of the WW domain of the E3 ubiquitin ligase Itch [119].

#### 3.3.4. Possible Regulation of Receptor Clustering by Phase Separation of the CT

As receptor oligomerization is crucial for ErbB activation, the regulation of their local concentration is of interest. Liquid-liquid phase separation (LLPS) is emerging as a major regulation mechanism in the cell, especially for IDPs. Can the sequence properties of CtErbB2 drive local LLPS, and clustering of the receptors in the membrane? Indeed, a high concentration of tyrosines, arginines [129] or glycines [130], as observed in CT-ErbBs (see Table 1), was shown to be associated with phase separation. Additionally, phase separation properties of tails were shown to affect clustering of several receptors [131].

In our NMR study of isolated CT-ErbB2 [30], we did not observe phase separation at room temperature and 200 mM salt concentration, even at hundreds of μM of protein. However, the multivalent properties of CT-ErbBs could lead to LLPS in the presence of adaptor proteins, as it was shown for the LAT receptors [132]. When this receptor is tyrosine-phosphorylated, binding of Grb2 and Sos causes clustering of the receptors. Such phenomenon was never, to the best of our knowledge, shown for ErbB receptors, but this remains to be fully investigated.

### 3.4. Structural Description Outside of Regions Interacting with the Kinase

A lot of structural information is available on the CT in the regions where it is in close interaction with the kinase domain, and is therefore seen in crystal structures. That is the case up to residue 1016 of EGFR. However, there is still about 200 residues with unknown conformation, except in very short segments in interaction shown in Figure 4. Few teams took up the challenge of studying such a long disorder segment. Moreover, full description of the tail conformation should include phosphorylated states, which are, to our knowledge, not well described in terms of multiplicity and heterogeneity, and difficult to reproduce. The phosphorylated states structurally described so far are therefore either phosphomimetics [28], synthetic monophosphorylations [29], or in vitro autophosphorylation products which were shown to have, on average, one phosphate per tail, which may not be physiologically relevant [34,35].

#### 3.4.1. Local Structure

Keppel et al. were the first ones to show experimentally that the tails of EGFR and ErbB3 are indeed disordered, using a wide variety of biophysical techniques [26]. They used the whole intracellular variant (including the kinase domain) to show that the first 50 N-terminal residues of the CT, which interact with the kinase, are more solvent-accessible than the kinase core itself, but still exhibit some solvent protection. This also holds true for ErbB2, which they included in this experiment. Their circular dichroism (CD) results suggest a small amount of residual secondary structure, but they did not investigate this aspect further. Other attempts were made to interpret CD spectra of both autophosphorylated and unphosphorylated variants [28,29,35], but single CD spectra of IDPs are difficult to interpret quantitively [133] due to the transient nature of their secondary structure elements and overlooked polyproline type II (PPII) helices content. PPII helices are left-handed helices of 3 residues per turn, that are favored by high proline content but do not strictly require prolines [134]. They were shown to be relatively common in disordered proteins and unfolded states [135].

Our group investigated CT-ErbB2 in more depth at the local level in the unphosphorylated state using NMR [27,30]. We showed that isolated CT-ErbB2 had a transient AP-2 α helix (shown in Figure 2 for EGFR) even when not bound to the kinase domain. Additionally, the high proline content of CT-Erbb2 was correlated with the presence of several transient PPII helices. PPII helices are interaction motifs for SH3 domains, especially if they contain PxxP motifs [136]. Although several SH3-containing proteins, such as Grb2 or Fyn kinase, are involved in ErbB signaling pathways, direct interaction of SH3 domains with those receptors was, to our knowledge, never shown. It was hypothesized [137] but only shown with ErbB peptides in vitro [138]. This opens new questions on regulation of ErbBs and signal transduction through the direct interaction of ErbB receptors and SH3 domains in vivo.

Upon phosphorylation (autophosphorylation, selective phosphorylation of single sites, or phosphomimetic mutations), the CD spectrum of CT-EGFR was slightly modified, and suggests local and/or small degree of loss of structure [28,29,35].

#### 3.4.2. Compaction and Long-Range Contacts

In the unphosphorylated state, Keppel et al. showed that isolated CT-EGFR and CT-ErbB3 were rather extended, consistent with their disordered nature [26]. However, even without a folded core, some transient long-range contacts cannot be excluded. Interestingly, SAXS data with and without urea showed the clear presence of more compact structure in CT-ErbB3 compared to CT-EGFR.

The first long-range contact to be noted in CT-EGFR is the one detected with the kinase core. Fluorescence studies showed that this contact was partially disrupted upon autophosphorylation of the intracellular domain [34], which left the CT more flexible [35]. Looking more closely into long-range contacts within the CT, contrary to CT-kinase contacts, some contacts were formed within the tail only upon phosphorylation of EGFR [29,139]. We showed that in unphosphorylated CT-ErbB2 there was also a transient long-range contact [30].

The contacts identified within CT-EGFR and CT-ErbB2 both bring regions that are separated by around 200 residues closer. The exact nature of these contacts is not obvious from amino-acid sequence, and the absence of salt-concentration dependence excludes electrostatic interactions in EGFR [28]. Comparison between different ErbBs is also difficult due to large sequence divergence outside of the residues close to the kinase domain [39]. Whether these contacts are specific at all, as well as their influence on function, thus remains to be investigated.

### 3.5. Comparison with the Intracellular Tail of Other RTKs

#### 3.5.1. Inhibition and Activation Mechanisms

The whole activation mechanism of ErbBs is strikingly different from the prototypical mechanism for RTKs. Firstly, the ligand-induced dimerization of ErbBs is solely receptor-mediated, meaning that the ligands do not participate in the dimerization interface necessary for activation of the receptors [41,140]. Amongst RTKs, ErbBs share this feature only with insulin receptors. Furthermore, the activation of the kinase domain does not require phosphorylation of its activation loop or juxtamembrane region in a trans-autophosphorylation mechanism, as seen in many RTKs [65]. Instead, ErbB kinases are activated allosterically by asymmetric dimerization [62]. Is the role of the CT also unique? A first clue is the length of the CT, which is longer in ErbBs than in other RTKs, with the exception of the Lemur Tyrosine Kinase receptor (LMTK) family, which was shown since its initial classification as RTKs to not actually phosphorylate tyrosines but serine/threonines, and does not have extracellular ligand-binding domains [141].

The beginning of the CT (including the AP-2 helix region and the electrostatic hook) of ErbBs was shown to be well-conserved amongst ErbBs, to have co-evolved with regions of the kinase domain, and to have diverged from other RTKs [32], which is consistent with the unique regulation role of this region. Other RTKs have an autoinhibitory role of the C-terminal tail disrupted by autophosphorylation, like Tie2 [142].

#### 3.5.2. An Overlooked Role of Prolines in ErbBs and Other RTKs?

Studies of interaction motifs in CTs of RTKs have focused on phosphotyrosines. Could it be that other motifs were missed? The unusually high content of prolines in ErbBs tail (as pointed out in Table 1) has barely been the subject of any investigation. A PPAY motif was identified in ErbB4 for binding to the WW domain of a ubiquitin ligase [119]. Additionally, our study of ErbB2 [30] showed two hints of the importance of prolines: the presence of proline interaction motifs (PxxP motifs for interaction with SH3 domains) and that of polyproline II (PPII) helices, also indicating the possibility of interaction with SH3 or WW domains. In signaling pathways rich in these domains, the possibility of yet uninvestigated regulatory interactions can be raised. Observation of RTK CT sequences shows that virtually all of them contain proline-rich segments. In particular, ROR orphan receptors CTs contain about 9% of proline residues, which goes up to more than 14% for Anaplastic Lymphoma Kinase (ALK), Leukocyte Tyrosine Kinase (LTK) and Fibroblast Growth Factor (FGF) receptors, and even about 20% for LMTK pseudo-RTKs. Interestingly, many of these RTKs with the highest proline content do not have known ligands or lack extracellular ligand binding domains altogether. That applies to ErbB2, the orphan receptor of the ErbB family, which has an even higher proline content than other ErbBs. It remains an open question whether the proline content allows additional levels of regulation to compensate for the absence of ligand regulation.

The Ladbury laboratory has been focusing on FGFR receptors, which are not orphan receptors but still have high proline content in their tail. They showed that Grb2 C-terminal SH3 (CSH3) domain could bind proline-rich segments of FGFR2 [143], not only preventing dephosphorylation of the tail by Shp2 but also blocking FGFR2 dimer in a semi-active state: the kinase was active, but CT autophosphorylation and subsequent signaling were prevented [144,145]. The phospholipase Plcγ1 SH3 domain also bound to the proline-rich region of FGFR2, competing with Grb2 and adding a layer of signaling regulation [146]. Very recently, the proline motifs of CT FGFR2 were also linked to the ability of the CT to regulate kinase activity by binding to the kinase domain and juxtamembrane segment [147].

## 4. Outlook and Outstanding Questions

ErbBs are a beautiful example of how structural biology, with the successive crystallization of the different domains of the receptors, can help understand the molecular mechanism governing the function of a complex protein with multistep activation. CT-ErbBs are also a prototypical example of how IDRs were ignored in such studies for decades. CT-ErbBs have features expected from IDRs involved in signaling and regulation: they have multiple phosphorylation sites embedded in interaction motifs and a lot of partners that can bind to different motifs or compete for the same one.

Biochemical and structural studies of CT-ErbBs also highlight specificities that are still overlooked in IDPs and IDRs, and are still very open questions that the studies reviewed here only started to answer: the role of prolines on conformation; the role of LLPS phenomena in receptor clustering; the regulation of (multiple) tyrosine phosphorylation and its conformational impact; the interaction between IDRs and folded domains of the same protein; and finally, the direct or indirect interaction (structurally and dynamically) between different motifs of the IDR, be it for post-translational modification or interaction with partners, which can only be studied by using full-length protein variants and not short peptides. These new questions are condensed in the summary Figure 5. A lot of work is still needed to understand all the regulation phenomena at play. In particular, experimental or in silico description of the phosphorylated states of the CT and studies of the interplay between the multiple interactions with partners in those different states are very much needed.

These questions apply to ErbBs, other RTKs, but also other types of receptors. More and more publications studying intrinsic disorder in receptors were published recently, for example on the N-methyl-D-aspartate (NMDA) receptor [148], class 1 cytokine receptors [149], or T-cell receptors [150]. Tackling these challenges requires new methodologies, in structural biology as well as in sample preparation and simulations, and could open new interesting perspectives in the development of drugs targeting IDPs and IDRs.

## Figures and Tables

**Figure 1 biomolecules-11-01690-f001:**
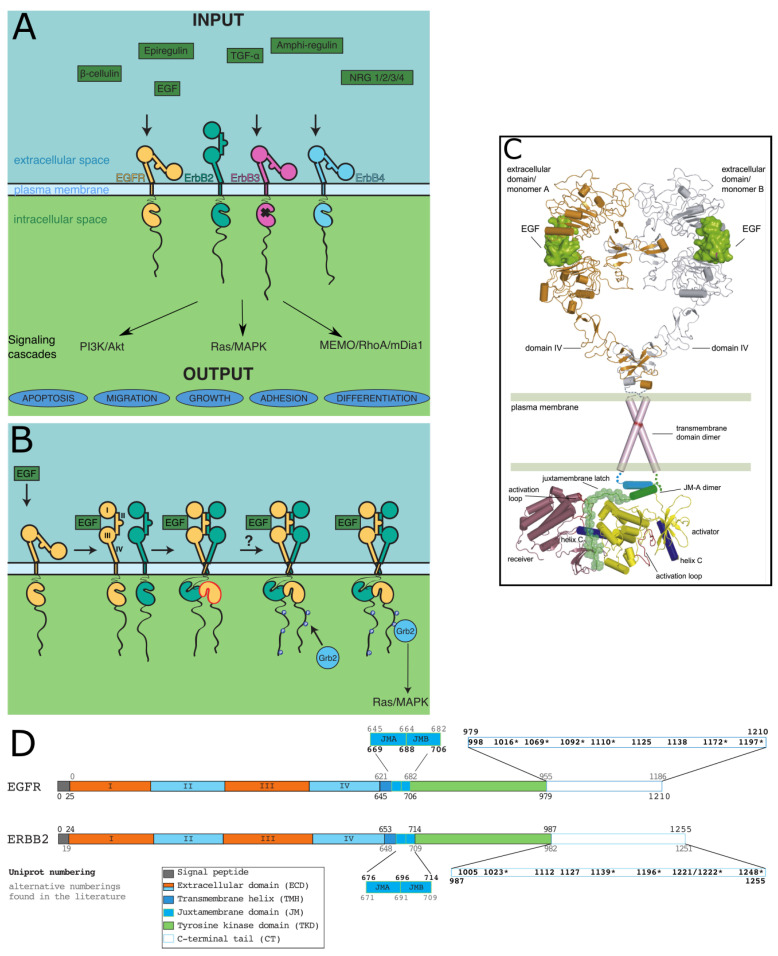
ErbBs structure and signaling mechanism. (**A**) Schematic representation of the role of ErbBs in signal transduction. Only some of the signaling pathways are represented here. The cross on ErbB3 kinase domain indicates its weak activity. ErbB2 is shown in its constitutively dimerization-prone conformation. (**B**) The different steps of ErbB signal transduction, with the example of ErbB1 ligand EGF triggering ErbB1/2 heterodimerization, ErbB1 kinase activation, and Ras/MAPK pathway activation through binding of the adaptor protein Grb2. (**C**) Model structure of activated full-length EGFR from structures and models of individual domains. A model for the C-terminal tail (CT) is lacking. Reproduced with permission from [31]. (**D**) Overall domain organization of EGFR and ErbB2 with numbering. For the ECD, in orange are leucine-rich repeat domains (domains I and III), and in blue are cystein-rich domains (domains II and IV). The Uniprot numbering is used (unless otherwise stated) in this review. For EGFR, the alternative numbering sometimes used excludes the signal peptide in the N-terminus. For ErbB2, the alternative numbering is that of rat ErbB2 (neu). In the CT, all tyrosine positions are indicated, with * on the ones that are known autophosphorylation sites.

**Figure 2 biomolecules-11-01690-f002:**
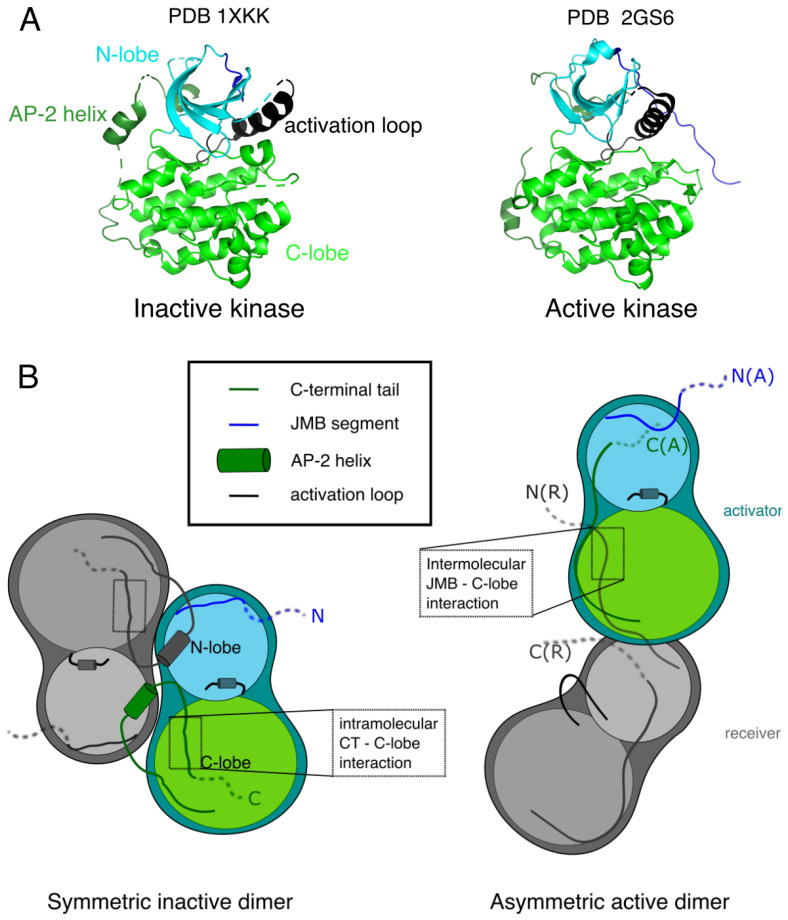
Inactive-to-active transition of ErbB kinase dimer and role of C-terminal tail. (**A**) Kinase monomers in inactive and active states, with change seen in activation loop conformation (black). Figure created using PyMOL (PyMOL Molecular Graphics System, Version 2.0, Schrödinger, LLC, New York, NY, USA). (**B**) Inactive and active kinase dimers. In the inactive dimer [31] (PDB 3GT8), the C-terminal tail interacts with the C-lobe of the same monomer (interaction region shown in a dotted rectangle). In the active dimer [62] (PDB 2GS2), the same region of the C-lobe of the activator kinase monomer interacts with the JMB segment of the receiver kinase monomer.

**Figure 3 biomolecules-11-01690-f003:**
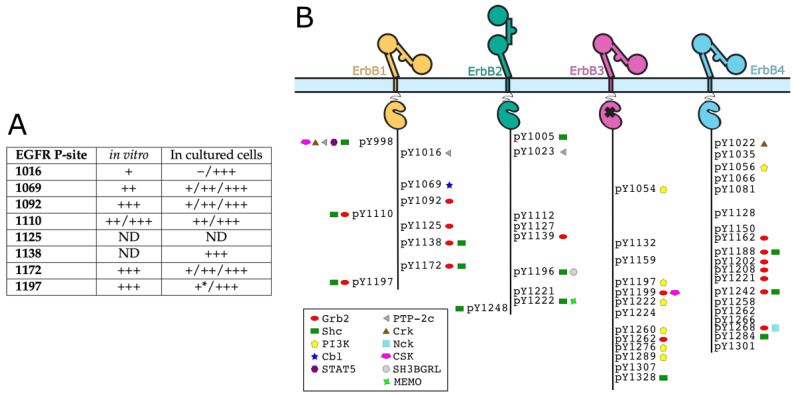
Phosphorylated tyrosines of ErbBs. (**A**) Review of different studies on relative levels of phosphorylation of 8 (out of 9) tyrosines of EGFR C-terminal tail. —, +, ++ and +++: different levels of phosphorylation, from low to high. *: greatly enhanced by ErbB2 expression. “/” indicates different results in different studies. “ND”, not determined. Adapted from [79]. (**B**) Phosphotyrosine interactome of C-terminal tails of ErbB receptors. Adapted from [84].

**Figure 4 biomolecules-11-01690-f004:**
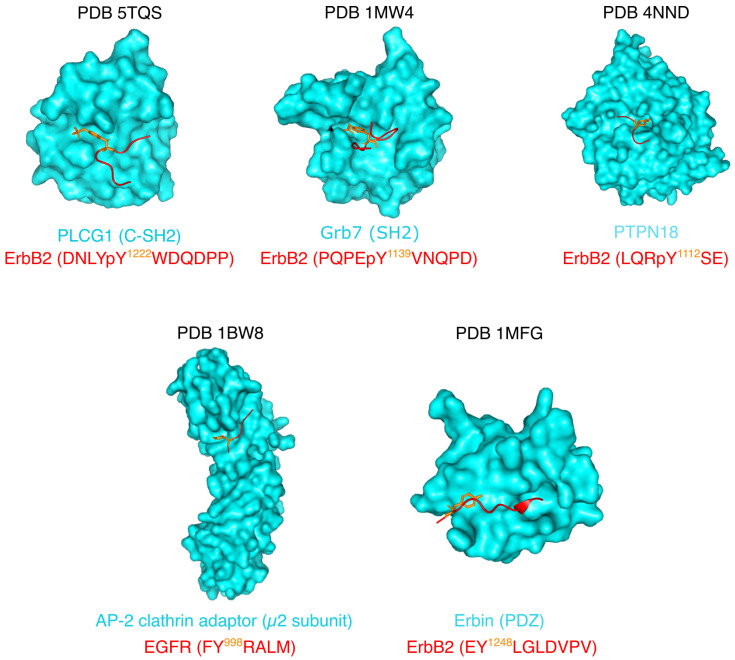
Experimental structures of CT-ErbB peptides (in red) bound to partners (in blue). Each (phospho)tyrosine mainly involved in binding is in orange and its number in the sequence is indicated. Figure created using PyMOL.

**Figure 5 biomolecules-11-01690-f005:**
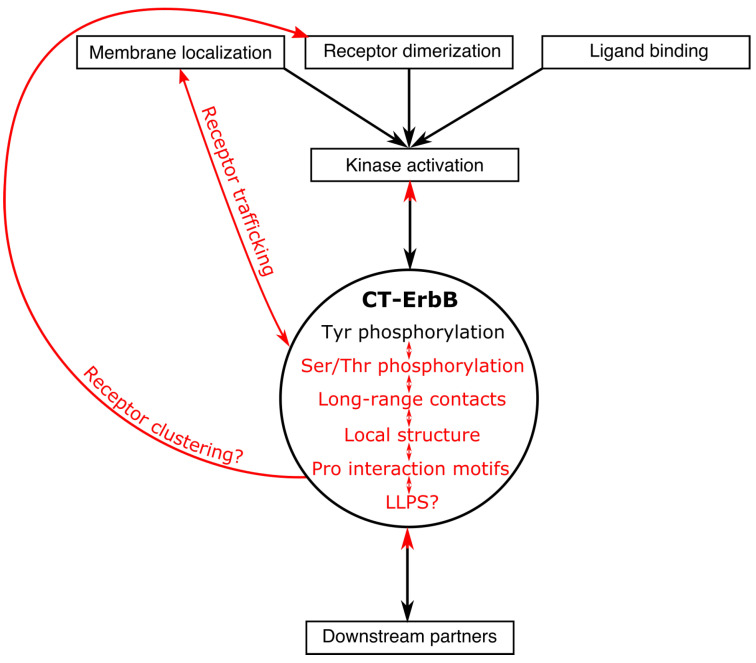
Summary of all experimentally confirmed or potential regulation roles of C-terminal tails of ErbBs (red), in addition to “canonical” roles of the CT (black). Each CT feature (in the main circle) can influence other ones and create regulation loops.

**Table 1 biomolecules-11-01690-t001:** Comparison of some sequence characteristics of CT-ErbBs. The C-terminus of each receptor is defined as the part C-terminal of the kinase domain as defined by Uniprot (ErbB1: P00533; ErbB2: P04626; ErbB3: P21860; ErbB4: Q15303). IUPred2A was used [99] (https://iupred2a.elte.hu/, accessed on 26 June 2021). ND = not determined.

	CT-ErbB1	CT-ErbB2	CT-ErbB3	CT-ErbB4	Average in Disprot [100]	Average in SwissProt [100]
**Length (residues)**	231	268	376	323	ND	ND
**Average IUPred score**	0.58	0.75	0.72	0.59	ND	ND
**Number [%] of Tyr**	9 [3.90]	9 [3.36]	14 [3.72]	19 [5.88]	[2.13]	[3.03]
**Number [%] of Gly**	10 [4.3]	28 [10.4]	34 [9.0]	15 [4.6]	[7.4]	[7.0]
**Number [%] of Pro**	24 [10.4]	44 [16.4]	39 [10.4]	39 [12.1]	[8.1]	[4.8]
**Number [%] of Asp**	20 [8.7]	21 [7.8]	19 [5.1]	21 [6.5]	[5.8]	[5.4]
**Number [%] of Glu**	12 [5.2]	19 [7.1]	38 [10.1]	27 [8.4]	[9.9]	[6.7]
**Number [%] of Arg**	8 [3.5]	11 [4.1]	29 [7.7]	18 [5.6]	[4.8]	[5.4]
**Number [%] of Lys**	6 [2.60]	5 [1.87]	7 [1.86]	15 [4.64]	[7.85]	[5.92]
**Total net charge**	−18	−24	−21	−15	ND	ND
**pI**	4.43	4.29	5.05	4.91	ND	ND

## Data Availability

Not applicable.

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
