# Peer review of "Expanding the Disorder-Function Paradigm in the C-Terminal Tails of Erbbs"

_biomolecules, 2021, doi:10.3390/biom11111690_

Round 1
Reviewer 1 Report
The review by Pinet and coworkers on the structure-function-disorder relationship in ErbB receptors is a very nice, well-written and comprehensive overview of the work done so far on the group of 4 tyrosine kinase (RTK) receptors and provides the view that the disordered C-terminal tails constitute more than passive interaction and scaffolding domains, and play active regulatory roles in signaling. This view is an emerging view broadly in membrane proteins and thus also timely to put forward for this group of proteins. The review puts focus on a generally highly overlooked region of these receptors, which constitute rather long disordered parts of the receptors. The review will be a very important asset to the ErbB-community, but also with an additional paragraph at the end, generally to the broad fields of single-pass receptors, where disorder play various different roles. Although the manuscript is very comprehensive, I suggest the following additions that will make the review much more useful, also for others outside the ErbB field.
Most importantly, the figures need work. And – they are too sparse. Figure 1 is completely un-readable - the font sizes are too small, the font colors too light to read, and the text in the colored boxed impossible to read. Try printing it and see for yourself. Why are the A and the B box not of the same size? Same goes for Figure 3 – the lines are hardly seen and inter-and intramolecular interactions are not distinct. Maybe an additional illustration/sketch of the whole receptor shown from the side would be beneficial? It is very difficult to imagine how conformational changes upon ligand binding may be translated to the kinase domains and result in the conformational exchange between the symmetric inactive dimer to the asymmetric active dimer. From figure 3 alone it is very difficult to see where the JM region is and what this means for the position of the TMH. And, since there are so many, many very complete 3D structures available, at least include them when this makes sense, this is in particular when discussing signal translation through the membrane (TMH:TMH interaction), dimer formation and especially when describing the interaction with the CT and the kinase domains (p 9 – very much needed). A real structure would also help define the activation loop, N-lone C-lobe, JMA, JMB etc. These abbreviations and structure definitions are used throughout, but for a person outside the field, these are not necessarily implicit. These structures are available and will increase the understanding of the inhibitory interactions. Are there no structures available on any full-length receptor?
Also, since this concerns the CT, it would be appropriate to include a figure showing the structures of the complexes available where peptides from CT has been solved in complex with partners (p. 7). This would add a nice overview – in particular of how little is actually done.
In general, but especially in section 3.3.2, p. 9 references to figure(s) are missing (here figure 3).
Do these receptors undergo phase separation? The many tyrosines and arginines combined with the depletion in lysines could suggest they do. Or, will the presence of the prolines be somewhat inhibitory of that? A discussion on this should be included. And, is there any report on cross-talk between the receptors and the membrane/lipids as reported for other membrane proteins and IDRs? If not, is this then only because it has not been studied or/and is this absence relevant to the signaling of this class of receptors?
In the discussion of the degradation of the receptors, are there any (phosphor)degrons present? or other SLiMs? Will an elm.eu.org screen identify any? Or are there any known in the literature? In general, I think it would be very interesting if the authors had mapped some SLiMs or performed a SLiM prediction, as it may explain some of the sequence characteristics e.g. the high content of Pro that is discussed in section 3.1, p7. Focus is here entirely on tyrosines and tyrosine phosphorylation but what is known about ser/thr phosphorylations, which are often involved in receptor trafficking and degradation? Is there a reason why this is not discussed?
In the end of the review, the authors open up towards other RTKs and compares ErbBs to those, which is really appreciated, but as disorder is also present in many other single pass receptors that recruit tyrosine kinases, are the finding for ErbBs more generally applicable for these fields, or are the disorder in these systems playing other roles? Or, have the investigations of these disordered tails not yet been done? If so, this review could spur more work in these fields as well, as disorder prevail in many of these intracellular domains.
In the conclusion it is mentioned that targeting IDRs in ErbBs would be relevant. For this is would be appropriate if the review includes a mentioning of known disease-causing mutations in the CTs. How do they link to the described functions of the CT? The authors bring forward a number of outstanding questions which lift the review and it could be useful if these questions are summed in a box or figure for easy access, and if they could briefly touch upon these in the conclusion, which then could be renamed to Outlook and outstanding questions – this would be helpful not only to the ErbB field but to many other fields involving membrane proteins.
Minors
Some language issue impacts the reading which is a shame as many very useful points are presented. I have highlighted a few examples below, but there are more
Why is “active” in brackets in the abstract?
p.2 l. 24 - When listing the receptors, please provide the preferred name in bold (the name used in this review)
p.2 l 38 – have the TMHs been crystallized? The text indicates this
- 3 l 69 – it was determined -> it was found
- 3 l 72 – what is meant by “functional specificities” – different downstream effects?
- 4 l 111 – remove actually
- 5 l 114 – “ has ever been described to our knowledge” -> “has to our knowledge never been described”
- 5 l 139 “the inactive an EGFR” – “the inactive EGFR”
- 5 l 157 - remove really
- 6 l 174 “recapitulative table” – an overview table
- 6 figure legend to figure 2 * is not shown in figure, instead +’s are.
- 6 l 184 forward. There is something odd. The first sentence indicates that we will learn about other kinases that phosphorylate ErbBs, but the following text lists interaction partner to the phosphotyrosines. Please correct
- 7 l 102 – explains what the activation of MEMO does, not well-known
- 7 l. 213 – the sentence does not make sense, regulation itself meaning? Also, reference is missing?
- 8 Table 1 – move the tyrosine content to the other aminoacids, it is easily overlooked as the last row in the table. Move up as the first in front of the Gly content
- 8 l 238 – “has been shown to not be” -> “ has been shown not to be”
- 8 l 249 – “a significant study” – I would prefer using another description of the study as this indicate some statistical comparison between studies.
- 9 l 267– Unclear if the ”C-terminal tyrosine” refers to the most C-terminal tyrosine or just a tyrosine in the C-terminal
- 9 l 292-93 – “first 70 to 80 residues” – of what? Unclear?
- 10 l 308 – “ recycling of” –> “recycling or”
- 10 l 310 – “thought EGFR” -> “thought that EGFR”
- 11 l 362 – which new questions? Unclear
- 11 l 367-68 – correct to “both bring regions that are separated by around 200 residues closer”
Generally – construct is referring to DNA, for proteins use variants
Reviewer 2 Report
This interesting review points out on the role of the C-terminal tails in ErbBs. The manuscript is well organized and understandable to readers at large.
Just some minor points:
1) Lines 139-141. This sentence is not clear, please rephrase it.
2) Some labels in Figure 1 are too tiny
3) In Conclusions it would be useful to list some open questions on the role of CTs in ErbBs (structure, mechanism) that may be addressed in the future (e.g. by structural biology or molecular modelling).
Reviewer 3 Report
This is a very comprehensive and well-written review covering an interesting and highly relevant topic.
I have only a few very minor suggestions for improvements:
- Line 9 and 10: The "active" and "passive" could be without quotation marks
- In the second paragraph of the Introduction, Figure 1 could be quoted earlier (and Fig. 1 brought forward in the text) when discussing the various domains to provide a visual overview for readers not so familiar with the ERBBs. Also when mentioning the C-terminal tails for the first time, an approximate indication of their length ("around 270 amino acids long") would be useful as a guide to readers to show that these are actually quite long.
- Line 174: instread of "recapitulative figure", the term "summary figure" may be clearer to readers
- Line 197: spelling error: "biais" should be "biases"
- Table 1: "Number [%] of -" presumably refers to negatively charged residues. Maybe this could be changed to listing the numbers and percentages of aspartic and glutamic acid residues as two separate lines because that has been done for the positively charged amino acids. This way the data would be more consistent overall.
